# Biosynthesis and Secretion of Human Tissue Kallikrein in Transgenic *Chlamydomonas reinhardtii*

**DOI:** 10.3390/md16120493

**Published:** 2018-12-07

**Authors:** Jun Chen, Jinxia Wu, Qingyu Wu, Zhangli Hu

**Affiliations:** 1Guangdong Technology Research Center for Marine Algal Bioengineering, Guangdong Key Laboratory of Plant Epigenetic, College of Life Sciences and Oceanography, Shenzhen University, Shenzhen 518060, China; szchenjun111@163.com (J.C.); wujx@szu.edu.cn (J.W.); 2Shenzhen Key Laboratory of Marine Bioresource and Eco-environmental Science, Shenzhen Engineering Laboratory for Marine Algal Biotechnology, College of Life Science and Oceanography, Shenzhen University, Shenzhen 518060, China; qingyu@tsinghua.edu.cn

**Keywords:** *Chlamydomonas reinhardtii*, human kallikrein, signal peptide, secretory vector

## Abstract

The green alga *Chlamydomonas reinhardtii* was recently been shown to be an effective bio-manufacturing platform for the production of recombinant proteins. The advantage of using *C. reinhardtii* is that it is fast to grow, inexpensive to culture, and relatively safe. However, the expression of foreign proteins is always low and difficult to purify in *C. reinhardtii*. Human kallikrein has the potential to be developed into certain drugs, like insulin. Therefore, its biosynthesis is important to drug development. In this study, we synthesized the *sg* gene, a signal peptide sequence of alkaline phosphatase, and inserted it into a pH124 plasmid, which contains a *HSP70A-RBCS2* promoter and a *RBCS2* terminator. Then, we inserted the human kallikrein gene *klk1* behind the *sg* sequence to make a pHsgk124 vector. The pHsgk124 were transferred into a cell-wall deficient strain of *C. reinhardtii*, cc-503, by using the glass bead method. Southern blot analysis showed that *sg* and *klk1* were incorporated into genes of the transgenic *C. reinhardtii*. RT-PCR analysis showed that it had an active transcription and its expression increased three times under heat stress. Western blot analyses of proteins inside and outside cells (in the culture medium) showed that *klk1* was expressed in the cell and the resulting protein was secreted into medium. An enzyme activity assay showed that the recombinant protein had the ability to hydrolyze the specific substrate H-D-Val-Leu-Arg-Pna. In conclusion, we successfully bioengineered *C. reinhardtii* to produce and secrete human kallikrein protein, which has important biomedical implications.

## 1. Introduction

*Chlamydomonas reinhardtii* is a type of green algae grown in freshwater. Since it is relatively safe, easy to culture and inexpensive to maintain, it has become a preferred model organism for biotechnology and bioengineering studies. With a well-characterized genetic profile, it has become an important bio-platform to express and produce recombinant protein and antibodies [1,2,3,4,5,6]. Many exogenous genes have been inserted and expressed in *C. reinhardtii* successfully, and some have been planned in commercial production, such as biodegradable plastic poly-3-hydroxybutyrate [7] and the human antibody IgG1 [8]. In recent years, our group successfully expressed exogenous genes *ble* and *egfp* in the mitochondria of *C. reinhardtii* [9,10].

Although transgene technology is well developed for nuclear, chloroplast [11] and mitochondria genomes in *C. reinhardtii*, the low protein expression is a major hurdle for developing it into a commercially viable bio-platform. Current recombinant proteins made by *C. reinhardtii* are mainly expressed inside the cells. When harvesting proteins, one needs to go through complicated cell breaking, purification, and re-purification processes. This not only reduces the yield of recombinant protein, but also sometimes causes its structural change. Using signal peptides [12] to construct a secreting vector to express exogenous genes has the advantage of secreting recombinant protein outside the cells, which facilitates the target protein’s collection and purification. The most successful applications of signal peptide are in bacteria and yeast. For example, the human growth hormone (20 kDa hGH) gene has been successfully incorporated into the *E.coli* genome to produce a secreted product [13], human α1-antitrypsin has been produced and secreted by *Saccharomyces cerevisiae* using inulinase signal, and human insulin has been expressed and secreted by *Pichia pastoris* [14]. There are many reports that *C. reinhardtii* as a platform can produce recombinant protein, however, most recombinant proteins are accumulated intracellularly, but there are few reports in the literature about using *C. reinhardtii* to produce secretory recombinant protein [15]. We performed the signal peptide analysis of the *C. reinhardtii* PHOX sequence (GenBank accession number: xp_001703098.1) and predicted that this signal peptide can guide the synthesis of PHOX protein from the nucleus to the extracellular, which is the foundation of secretory recombinant protein production.

Human tissue kallikreins (htKs) are widely expressed in various tissues and are a secretory protein mainly from the pancreas and kidney [16]. The htKs are encoded by 15 similar steroid hormone-regulated genes and the main one is *klk1*. The htK encoded by *klk1* is a single-chain serine protease that is synthesized as a preproenzyme. It is then cleaved off the signal peptide to become a pro-enzyme to be secreted. The cleavage of pro-peptide makes it an active enzyme [17]. The primary activity of htK is to cleave low molecular weight kininogen to release lysyl-bradykinin (LBK). LBK binds to its receptors B1 and B2 in target tissues to mediate various physiological processes, such as blood pressure regulation, smooth muscle contraction, neural plasticity, pain induction, vascular permeability, vascular cell growth, electrolyte balancing, and inflammatory cascades [18,19]. LBK also has a role in pregnancy maintenance. Dysregulated htK is associated with various diseases, including cancer. Hence, htK is a valuable protein which has potential in the treatment of hypertension, diabetes, cancer and nephropathy.

In this study, we constructed a secretory vector carrying an exogenous *klk1* gene, inserted it into *C. reinhardtii*, analyzed the gene and protein expression in *C. reinhardtii*, and measured resulting htK in the culture medium.

## 2. Results

### 2.1. DNA Verification of Transgenic Algae

PCR-electrophoresis results, including a wild strain sample, three transgenic samples of *C. reinhardtii*, (named sgk1, sgk 2 and sgk3, respectively) and a negative control are shown in Figure 1. The transfected genome was named *sgk* (*klk1* genome with an upstream signal peptide genome). Transgenic cells showed extra 780-bp segments on the gel. This confirms that the *klk1* gene was incorporated into the cc-503 genome. A further total DNA Southern blot (Figure 2) shows that a band appeared just below 1584 bps in the transfected cells but not in the non-transfected cells. The theoretical DNA segment of *HSP70A-RBCS2-sg-klk1-RBCS2* contains 1511 bps. The experimental conditions may have slightly affected the band position, furthermore, a correct sequence have to show in Southern blot. Therefore, the band appearance near 1584 bps should confirm that the segment of *HSP70A-RBCS2-sg-klk1-RBCS2* was successfully incorporated into the cc-503 genome.

### 2.2. Western Blot Analysis of Expressed Protein

A Western blot assay was used to analyze proteins inside cells of currently transfected cc-503, previously constructed strain k6-7 (successfully expressed *klk1* but non-secretory), and non-transfected cc-503. The result is shown in Figure 3. A band at around 55 kDa appeared in all transfected cc-503 cells (sgk1, sgk2, sgk3) and k6-7 (used as a positive control) cells, but not in non-transfected cc-503 cells. The molecular weight was bigger than the theoretical 31 kDa, which may be due to the glycosylation of recombinant proteins.

Next, Western blot assay was again used to analyze proteins in the 50-time concentrated *C. reinhardtii* cell culture medium to detect excreted extracellular htK protein. The result is shown in Figure 4. In the culture medium of k6-7, the band at 55 kDa disappeared, confirming that without signal peptide, htK made from *klk1* expression cannot get out of the cells. In the culture medium of transfected cc-503 cells (sgk1, sgk2, sgk3), the band at 55 kDa still exists, indicating htK protein made from *klk1* was successfully secreted into the medium from cells and the existence of htK protein in the medium was not due to cell lysate after cell death.

### 2.3. Gene Expression and Enzyme Activity under Heat Stress

The changes of the *klk1* gene expression of the transfected cc-503 after heat stress are depicted in Figure 5. Sixty minutes after heat stimulation, the mRNA level reached a maximum, about three times higher than that before heat stimulation. Then, at 90 min, it drop to lower than level of 0 min and maintained that level from 90 min till 180 min.

The enzyme activity analysis of the same samples (Figure 6) shows that the initial catalytic activity of htK (resulting from *klk1* expression) was at 1800 nmol/mg·h. The activity decreased somewhat at 30 min possibly due to heat stress. It then reached a peak activity of 2400 nmol/mg·h at 90 min, possibly due to heat activated *HSP70A*, which in turn increased the mRNA expression and then the protein expression.

## 3. Discussion

*C. reinhardtii* belongs to green algae, the ancestor of land plants [20]. It exhibits a wide capacity in biosynthesis, which presents unique opportunities for commercial exploitation using bioengineering. Much research work was done to use *C. reinhardtii* as a bio-manufacturing platform to produce mammalian/human recombinant protein [1,3]. Most recombinant proteins, achieved by using *C. reinhardtii*’s gene expression systems, are accumulated intracellularly in algae cells [21]. The harvesting and purification of intracellular protein present a significant challenge. Therefore, more researchers are starting to focus on the secretory expression of recombinant proteins.

Rasala et al. recently achieved the expression and secretion of xylanase1 in *C. reinhardtii* by using a signal peptide [15]. In our study, we inserted a signal peptide gene (*sg*) upstream of the target exogenous gene and achieved the secretion of the recombinant protein htK. Therefore, we show that in principle it is possible to engineer *C. reinhardtii* to produce secretory human protein. Even though the production is still relatively low, we believe that through proper adjustment and bioengineering, we will be able to enhance the extracellular production of the target human recombinant protein. Interestingly, in Rasala and co-workers’ report, they found that extracellular and intracellular xylanase1, both produced by transgenic *C. reinhardtii* cells, had different migration speeds in the SDS-PAGE assay. The extracellular one moved slower than the intracellular one, even though they have very similar enzyme activities [15]. This phenomena suggests that the extracellular secreted protein may have gone through a post-translational modification or glycosylation. We also found similar a phenomenon which shows that the recombinant protein’s molecular weight was higher than the theoretical one, both intracellular and extracellular, even though the activities of the enzymes were similar. Further experiments are needed to fully elucidate the nature and process of this post-translational modification. In our design, we used endogenous signal peptides of *C. reinhardtii*, therefore, it should be easily recognizable by *C. reinhardtii*. In theory, linking this signal peptide with other exogenous genes to produce various secretory recombinant proteins should be feasible. We are in the process of testing this signal peptide on more exogenous recombinant proteins to verify this feasibility.

In 2000, Schroda et al. inserted *HSP70A* upstream of the *RBCS2* promoter to get an enhanced gene transcription and protein synthesis through light–heat stimulation [22]. In their report, heat stress significantly enhanced arylsulfatase (*ARS*) gene expression, its mRNA level reached a maximum after 60 min, and its enzyme activity reached a maximum at 90–180 min, 1000 times higher than that without stimulation. We inserted *HSP70A* upstream of our target genes and have observed the enhancement of the transcription after heat stimulation as well, albeit at a much lower level. Under heat stress, the mRNA of *klk1* was only enhanced three times (Figure 5) and enzyme activity only increased 33.3% (Figure 6). This may be due to the fact that *klk1* is an exogenous gene to *C. reinhardtii*. Apart from inserting *HSP70A* and applying heat stress, other strategies to enhance production may also be considered, which include using priority codon and inserting regulation genes.

## 4. Materials and Methods

### 4.1. Construction of Plasmid

SignalP V3.0 (http://www.cbs.dtu.dk/services/SignalP/) was used to predict phosphate-repressible alkaline phosphatase (GenBank entry: XP_001703098.1) signal peptide and the signal peptide gene *sg* was synthesized accordingly. The vector used was pH124 (kept in our laboratory), which contains the *HSP70A-RBCS2* promoter and the *RBCS2* terminator [6]. The *sg* was inserted to obtain a secretory pHsg124 vector. Then, the human *klk1* was inserted downstream of *sg* to obtain the pHsgk124 vector (see Figure 7).

The glass-bead method [6] was used to transfect *HSP70A-RBCS2-sg-KLK1-RBCS2* into *C. reinhardtii* cc-503. Transfected *C. reinhardtii* cells were cultured in plates containing 10 µg/mL Zeocin.

### 4.2. Materials and Culture Conditions

The cell wall deficient strain cc-503 of *C. reinhardtii* was purchased from the *Chlamydomonas* Center at Duke University (Durham, NC, USA). K6-7 is a transgenic strain of *C. reinhardtii* developed by our laboratory previously, which does not have a signal peptide. A tris-acetate-phosphate medium was used to culture algae and the algal cells were under continuous irradiation for 24 h a day at 22 °C.

### 4.3. PCR Analysis

Based on known Chinese *klk1* (GenBank entry: AY703451), primers were designed as the following:

Prsklk11: 5′-AAGATCTTCATGTGGTTCCTGGTTCTGTG-3′

Prsklk12: 5′-GCCACGTGACGCGTTCAGGAGTTCTCCGCTATGGT-3′

The PCR analysis for sequence confirmation was done by sending synthesized samples to Takara Biotechnology Inc. (Dalian, China) for sequencing. The gel running conditions were: 94 °C for 10 min, followed by 35 cycles of 94 °C for 60 s, 60 °C for 60 s, and 72 °C for 60 s, plus a final extension for 10 min.

### 4.4. Total DNA Southern Blot

Total DNA extraction was performed using DNeasy kit (Takara Biotechnology, Dalian, China) and following the Roche’s (Mannheim, Germany) manual and digoxin (DIG) high prime DNA labeling and detection starter kit I, as per the manufacturer’s instruction.

### 4.5. Fluorescence Quantitative PCR

*C. reinhardtii* was cultured under normal conditions (in a conical flask at 22 °C with a light intensity of 25 μmol.m^−2^·s^−1^) up to the logarithmic phase. Just before heat stimulation a 1 mL sample was taken. The culture was then moved to a 40 °C incubator, heat-stimulated for 30 min, and another 1 mL sample was taken. The culture was moved back to normal culture conditions and 1 mL repeated sampling was done every 30 min five times till reaching the 180 min mark. Each sample was immediately centrifuged at 13,800 *g* for 1 min at 4 °C. Then, the supernatant was removed and the remaining sample stored under −80 °C until analysis.

The total RNA extraction was done on those samples collected at the above mentioned seven time points. The RNA extraction was performed using the RNAfast200 kit (Fastagen, Shanghai, China). A quantitative fluorescence PCR machine (ABI 7300, ThermoFisher, Waltham, MA, USA) was used to measure the RNA. Each sample was diluted to 250 mg/mL before a reverse transcription using Takara RNA PCR kit (V. 3.0, Takara Biotechnology, Dalian, China) was performed according to the manufacturer’s instructions. The primers used for *klk1* were:

Prklk1rt1: 5′-TGGGTGCTCACAGCTGCTCA-3′

Prklk1rt2: 5′-GTGTGGGAAGCTCTCACTGA-3′

The primers used for *C. reinhardtii β-actin* were:

Practin1: 5′-ACCCCGTGCTGCTGACTG-3′

Practin2: 5′-ACGTTGAAGGTCTCGAACA-3′

### 4.6. Protein Western Blot Analysis and Enzyme Activity Measurement

*C. reinhardtii* cell protein of 10 µg and 10 µg of super concentrated (50 times) cell culture medium were used for Western blot analysis. htK monoclonal antibody was used as the primary antibody and alkaline phosphatase-labeled sheep anti-mouse IgG was used as the secondary antibody (both purchased from Sigma-Aldrich, St Louis, MO, USA). The htK was reduced by thermolysin (4 units). Then, the htK enzyme activity was measured using a photometer, which measures the transformation of the substrate D-Val-Leu-Arg-PNA (S-2266, Chromogenix, Milano, Italy) by htK at 405 nm. Enzyme activity was standardized to the total protein of the respective sample. The protein concentrations were determined by using the Bradford assay.

### 4.7. Statistical Analysis

Each experiment was performed in triplicate and results were presented as mean ± SD. One-way ANOVA analysis was performed with Graphpad Prism 5. Significant differences were defined as *p* < 0.05 via Dunnett’s post-hoc test.

## 5. Conclusions

We designed a signal peptide and resulting secretory plasmid and incorporated it into *C. reinhardtii*. We successfully produced secretory human htK using transgenic *C. reinhardtii*. This technology can be used to produce large amounts of htK at a much lower cost than current bio-production strategy, and has significant medical and economic implications. This technology can also be used to produce other recombinant proteins that are important for medicine or industry.

## Figures and Tables

**Figure 1 marinedrugs-16-00493-f001:**
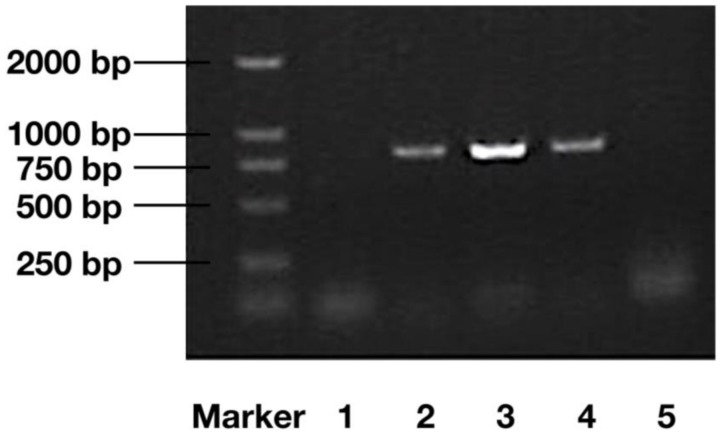
PCR-electrophoresis result for the *sgk* genome (*klk1* genome with upstream signal peptide genome) detection. Marker, DL-2000; 1, non-transfected cc-503; 2, sgk-1; 3, sgk-2; 4, sgk-3; 5, negative control (water).

**Figure 2 marinedrugs-16-00493-f002:**
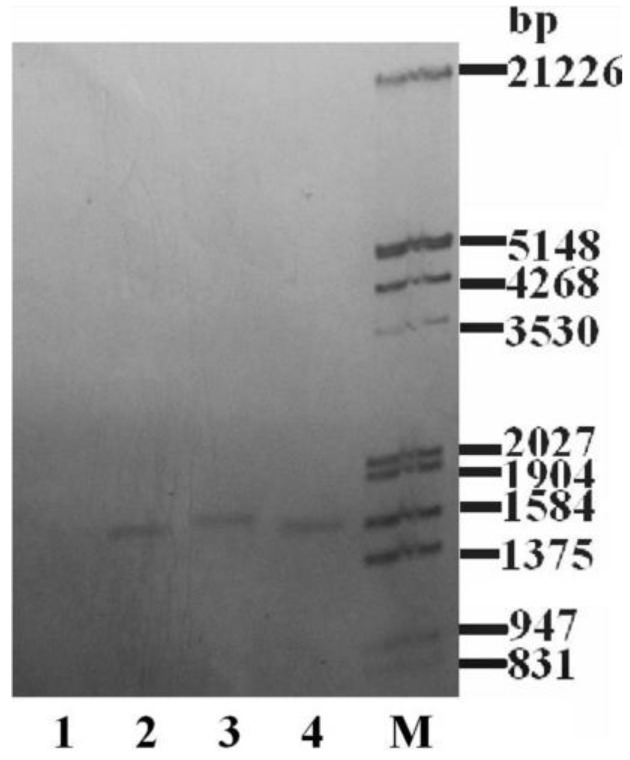
Southern blot analysis of transgenic algae with the *sgk* genome (using DIG high prime DNA labeling and detection starter kit I, see Section 4.4). M, marker; 1, non-transfected cc-503 DNA fragment; 2, sgk-1 DNA fragment; 3, sgk-2 DNA fragment; 4, sgk-3 DNA fragment.

**Figure 3 marinedrugs-16-00493-f003:**
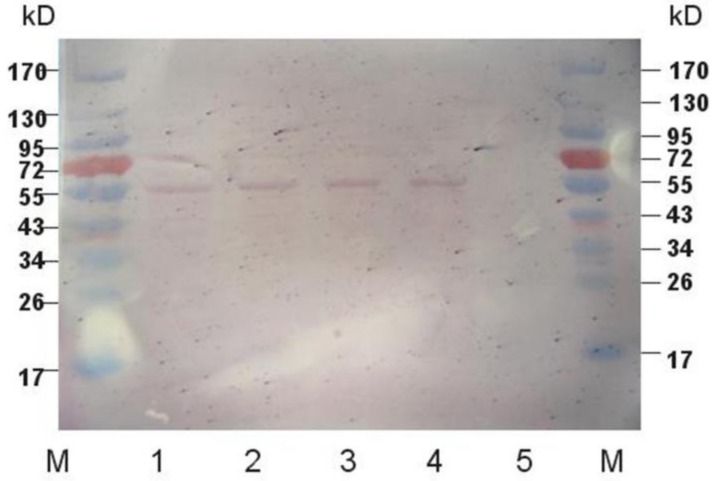
Western blot analysis of intracellular protein produced by transgenic and wild type algae. Marker, Fermantas PageRuler^TM^ (Glen Burnie, MD, USA) pre-stained protein ladder; 1, sgk-1; 2, sgk-2; 3, sgk-3; 4, k6-7; 5, cc-503.

**Figure 4 marinedrugs-16-00493-f004:**
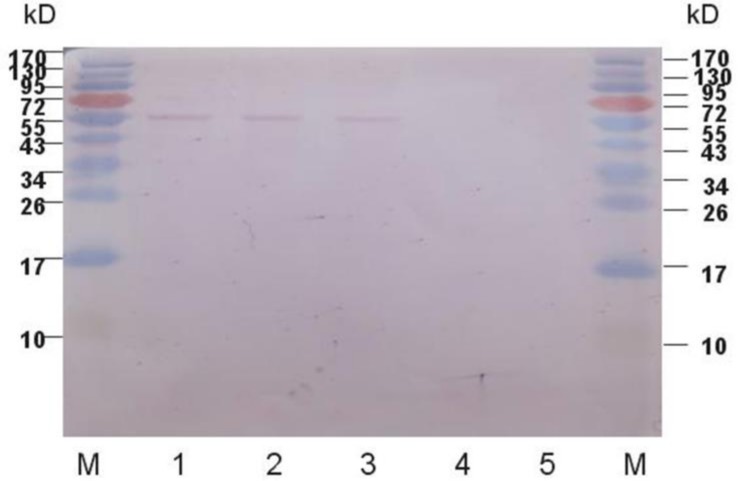
Western blot analysis of extracellular protein produced by transgenic and wild type algae. Marker, Fermantas PageRuler^TM^ (Glen Burnie, MD, USA) pre-stained protein ladder; 1, sgk-1; 2, sgk-2; 3, sgk-3; 4, k6-7; 5, cc-503.

**Figure 5 marinedrugs-16-00493-f005:**
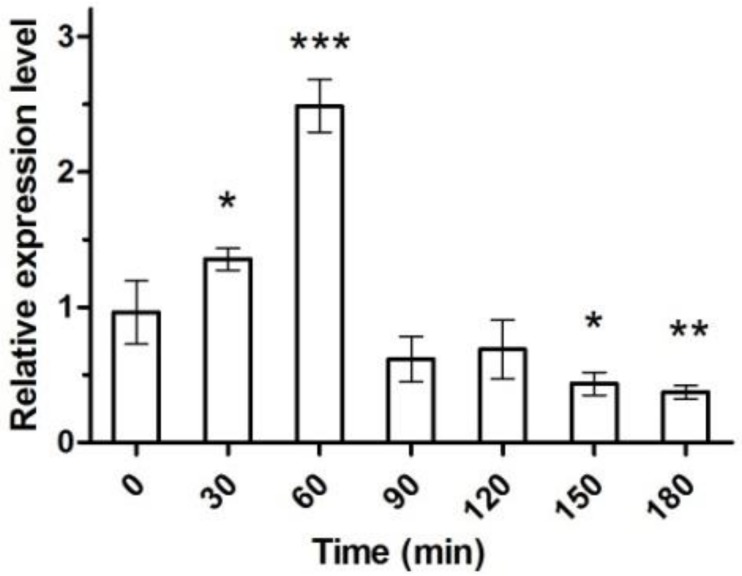
mRNA expression level of *klk1* in transgenic *C. reinhardtii* (sgk) cells under heat shock and intense light conditions. The relative transcription levels of *klk1* were determined after 0, 30, 60, 90, 120, 150, and 180 min by qRTPCR. β-actin was used as a reference gene and the values were normalized to the transcript levels in the control. Data are averages of biological triplicate, and the error bars represent standard deviation. One asterisk, *p* < 0.05; two asterisks, *p* < 0.01; three asterisks, *p* < 0.001.

**Figure 6 marinedrugs-16-00493-f006:**
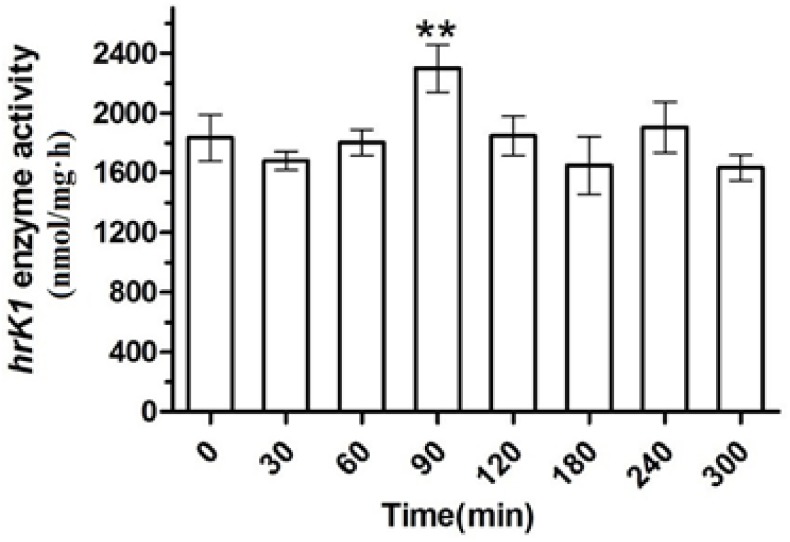
Enzyme activity changes of hrK in transgenic *C. reinhardtii* (sgk) cells under heat shock and intense light conditions. Significant difference versus 0 min time point sample was indicated with one asterisk (*p* < 0.05) and two asterisks (*p* < 0.01). Data were reported as mean ± SD of three independent repeated experiments.

**Figure 7 marinedrugs-16-00493-f007:**
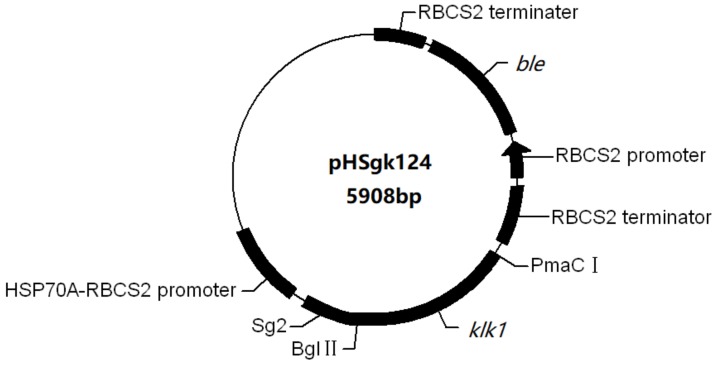
Construction of pHSgk124 plasmid. Abbreviations: *sg*, signal peptide; *klk1*, human tissue *kallikrein 1* gene; *ble*, bleomycin resistance gene; PmaC I, enzyme cleavage point 1; Bgl II, enzyme cleavage point 2.

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
