# Peer review of "Biosynthesis and Secretion of Human Tissue Kallikrein in Transgenic Chlamydomonas reinhardtii"

_marinedrugs, 2018, doi:10.3390/md16120493_

Reviewer 1 Report

This manuscript by Chen et al. describes the biosynthesis and the secretion of human tissue kallikrein using the green algae Chlamydomonas reinhardtii expression system. The authors cloned the gene with a signal peptide sequence, thus allowing secretion of the expressed protein into the culture medium. Owing the generally low expression level of heterologous proteins in this expression system and to the harsh conditions that would be required to lyse the cells and recover a heterologous protein expressed intracellularly, their secretion approach allowed the production and the recovery of significant amounts of pure and active protein. The authors conclude that the same approach and the same fast to grow, inexpensive and safe expression platform can be used to produce other recombinant proteins for a number of research and industrial applications.

Overall, this is an informative and well written paper. The results are clear and the experiments are described in details. Moreover, the content of the paper fits well with the topics covered by the journal. It will therefore be of interest for the readership of Marine drugs.

Minor:

While this manuscript is generally well written, some minor editing is still required prior to publication. For instance:

Lines 17-18.. “However…”. This phrase should be re-written in standard English

Line 73-74: “PCR…”. This phrase should be re-written in standard English

Line 103: “…secreted into medium…” should become “…secreted into the medium…”

Line 120: …Due to the heat stress” should become… “…due to heat stress” (i.e. remove “the”)

Line 122: …”then the protein expression. Should become “…then protein expression”. (i.e. remove “the”)

Line 127: “Data were averages…” should become “Data are averages…

Line 128: “…represented…” should become “…represent…”

Line 144: …by using signal peptide.” Should become “… by using a signal peptide. The same also in the same line…”we inserted signal peptide…” should become “we inserted a signal peptide…”

Line 168: “This may be due to that…” should become “This may be due to the fact that…”

My overall recommendation: accept after minor revision.

Author Response

Point 1: Lines 17-18.. “However…”. This phrase should be re-written in standard English

Response 1: Edited

Point 2: Line 73-74: “PCR…”. This phrase should be re-written in standard English

Response 1: Edited

Point 3: Line 103: “…secreted into medium…” should become “…secreted into the medium…”

Response 1: Edited

Point 4: Line 120: …Due to the heat stress” should become… “…due to heat stress” (i.e. remove “the”)

Response 1: Edited

Point 5: Line 122: …”then the protein expression. Should become “…then protein expression”. (i.e. remove “the”)

Response 1: Edited

Point 6: Line 127: “Data were averages…” should become “Data are averages…

Response 1: Edited

Point 7: Line 128: “…represented…” should become “…represent…”

Response 1: Edited

Point 8: Line 144: …by using signal peptide.” Should become “… by using a signal peptide. The same also in the same line…”we inserted signal peptide…” should become “we inserted a signal peptide…”

Response 1: Edited

Point 9: Line 168: “This may be due to that…” should become “This may be due to the fact that…

Response 1: Edited

Reviewer 2 Report

In this study, the authors synthesize of the sg gene, a signal peptide sequence of alkaline phosphatase, and inserted it into pH124 plasmid, which contains HSP70A-RBCS2 promoter and RBCS2 terminator. Then, they inserted human kallikrein gene klk1 behind the sg sequence to make pHsgk124 vector. The pHsgk124 were transferred into cell-wall deficient strain of C. reinhardtii, cc-503, by using the glass bead method. In conclusion, the authors successfully bioengineered C. reinhardtii to produce and secrete human kallikrein protein, which has important biomedical implication. This article should be published.

Author Response

Thank you for your review work

Reviewer 3 Report

1- The introduction could be improved, and adding information about signal peptide gene sg for reader is needed.

2- line 54-55 in the introduction is contradicts line 138-139 in the discussion, please re-write this.

3- lines 118-122, explain the result in figure 6, but the text does not match the bar chart. 

  A- The initial catalytic activity doesnt show at 1800 nmol/mg·h

  B- The activity did not decreased at 30 mints as mentioned in the text.

  C- Then it reached peak activity of 2400 nmol/mg·h 121 at 90 min, however, on the bar chart        the highest peak is at 60 min.

  D- Figures 5 &6 look similar, Please check them and better present these results.

4- Line 168 in the discussion, more justification needed on the enzyme activity as this part is very important.

Author Response

Point 1: The introduction could be improved, and adding information about signal peptide gene sg for reader is needed.

Response 1: The sg sequence is based on the amino acid sequence of the C. reinhardtii PHOX protein, GenBank accession number: xp_001703098.1, and the signal peptide analysis software SignalP V3.0 is used to predict the signal peptide which can guide the synthesis of the protein from the nucleus to the extracellular. We edited the introduction part and added this information for reader.

Point 2: line 54-55 in the introduction is contradicts line 138-139 in the discussion, please re-write this.

Response 2: Edited

Point 3: lines 118-122, explain the result in figure 6, but the text does not match the bar chart.

   A- The initial catalytic activity doesnt show at 1800 nmol/mg·h

   B- The activity did not decreased at 30 mints as mentioned in the text.

   C- Then it reached peak activity of 2400 nmol/mg·h 121 at 90 min, however, on the bar chart             the highest peak is at 60 min.

   D- Figures 5 &6 look similar, Please check them and better present these results.

Response 3: Edited

Point 4: Line 168 in the discussion, more justification needed on the enzyme activity as this part is very important.

Response 4: In this article, our main research object is signal peptide which can guide secretion of recombinant protein. Using the foreign protein klk1 of Chlamydomonas, we can detect whether this signal peptide can successfully guide the secretion of proteins from the nucleus to the outside of the cell by WB test. We successfully identified this model as a universal carrier for extracellular secretion. More experiment is in progress to improve the efficiency of this model, and more analysis of the activity of the enzyme we will discuss in the following experiments.
